# A Computational Framework for AI-Driven Generation and Psychometric-Based Personalization of Educational Content

## Abstract

This paper presents and evaluates a methodological blueprint for an adaptive learning system that integrates LLM-generated content with Item Response Theory (IRT). We conducted a large-scale simulation (N=10,000) where an agent administered a 30-item AP Chemistry question bank to virtual students, personalizing the question sequence based on real-time ability estimates. Our IRT-personalized agent achieved a statistically significant and robust performance gain over both random selection and fixed-difficulty baselines ($p < 0.0001$). A detailed subgroup analysis revealed that this performance lift extended across all learner ability levels and was most pronounced for students in the lower-to-average ability quartiles. Our findings validate the effectiveness of this synergistic framework and provide a strong, reproducible baseline justifying future human-subject experiments.

## 1 Introduction

Advanced Placement (AP) Chemistry is a cornerstone of high school STEM education, designed to be the equivalent of a first-year college chemistry course. Its curriculum is extensive, covering abstract concepts such as quantum mechanics, thermodynamics, and reaction kinetics. The inherent difficulty and breadth of the material often lead to varied learning outcomes among students (Tai et al., 2005). Traditional pedagogical methods, including lectures and uniform homework assignments, struggle to cater to the individual learning trajectories of each student. This heterogeneity necessitates a shift towards personalized learning paradigms that can provide targeted support (Corbett & Anderson, 1994).

The promise of personalized learning lies in its ability to adapt content, pace, and difficulty to a student's specific needs, a concept long explored in intelligent tutoring systems (ITS). Research has shown that well-designed ITS can produce learning gains equivalent to one-on-one human tutoring (VanLehn, 2011). However, the practical implementation of such systems has historically been hindered by the immense effort required to create and calibrate a vast repository of educational content. The advent of powerful Large Language Models (LLMs) has fundamentally altered this landscape (Brown et al., 2020; Touvron et al., 2023). These models can generate human-like text, making it feasible to produce a large volume of educational questions on specialized topics, a task previously requiring significant human effort (Polyak et al., 2023).

While LLMs solve the content generation problem, they do not inherently provide a mechanism for personalization. For this, we turn to the field of psychometrics. Item Response Theory (IRT) is a powerful statistical framework used to model the relationship between an individual's latent ability (e.g., knowledge of chemistry), the characteristics of an item (e.g., a question's difficulty), and the probability of a specific response (Lord, 1980; Hambleton et al., 1991). By understanding these relationships, an adaptive system can intelligently select questions that are optimally challenging for

a given student—neither too easy to be uninformative nor too hard to be discouraging. This principle is the foundation of modern Computerized Adaptive Testing (CAT).

This paper introduces and evaluates the computational framework that synergizes these two technologies. Rather than presenting a large-scale empirical validation, our primary contribution is the methodological blueprint for an end-to-end system that handles AI-driven content creation, psychometric seeding, and adaptive delivery. We demonstrate the framework's potential through a large-scale, controlled simulation, providing a vital proof-of-concept and a clear, reproducible baseline for future work. Our contributions are therefore:

1. A methodology for using LLMs to create a domain-specific question bank and seed it with initial psychometric parameters.

2. The design of a reproducible simulated learning environment to serve as a testbed for adaptive algorithms.

3. A large-scale computational experiment demonstrating that the framework produces statistically significant and robust performance gains across all learner subgroups, thereby justifying the next steps toward real-world pilot studies.

## 2 Related Work

The integration of Large Language Models (LLMs) and Item Response Theory (IRT) for personalized education exists within a rapidly evolving body of research. Our work builds upon several key areas: AI-driven question generation, the psychometric analysis of AI-generated content, and adaptive learning systems.

Recent studies have focused intensely on evaluating the quality of LLM-generated educational items. For instance, Luo et al. (2022) conducted a comparative study between AI-generated and human expert-designed vocabulary tests, providing a benchmark for quality assessment. Similarly, Laupichler et al. (2024) compared ChatGPT-generated medical exam questions to human-authored ones, highlighting cross-domain challenges and opportunities in ensuring pedagogical value. Our work extends this by not only generating questions but also integrating them into a dynamic, adaptive system.

The application of psychometric models to AI-generated content is a critical and emerging field. Young et al. (2025) performed a direct IRT analysis of multiple-choice items generated by ChatGPT-4, which is central to our contribution. Their work validates the feasibility of applying such models. Furthermore, Strugatski & Alexandron (2024) have explored using IRT to differentiate between human and AI responses, suggesting a novel role for IRT in quality assurance pipelines for AI-generated assessments. Our framework operationalizes these ideas by using IRT not just for analysis, but for personalized item selection in real-time.

Within the specific domain of chemistry education, researchers have explored the potential and pitfalls of AI. Clark (2023) investigated the use of an AI chatbot for answering general chemistry questions, while Humphry & Fuller (2023) examined broader applications in undergraduate labs. Emenike & Emenike (2023) raised important considerations regarding AI text generation for chemistry educators, emphasizing the need for expert oversight—a principle we incorporated through our human-in-the-loop verification step.

### 2.1 Comparison to Contemporary Student Models

Our framework's use of Item Response Theory (IRT) is a deliberate choice, situated within a broader landscape of student modeling techniques. Foundational approaches such as Knowledge Tracing (4) and its more recent Bayesian and deep learning-based extensions offer granular, concept-level tracking of student mastery over time. These models can be highly predictive in data-rich environments.

However, for the goals of this study—to provide a robust methodological blueprint for a new domain—IRT presents several key advantages. First, its parameters for item difficulty ($b$) and discrimination ($a$) offer direct psychometric interpretability, which is invaluable for content creators and educators (6). Second, IRT provides a parsimonious and computationally efficient model of overall student ability, making it highly effective in the "cold start" phase where large-scale interaction

data is not yet available. Our approach aligns with other contemporary research that validates the application of IRT to analyze and deploy AI-generated educational content (16; 12). While a full comparative study was outside the scope of this work, a critical next step is to benchmark our IRT-based agent against these state-of-the-art neural and knowledge tracing models as part of the planned human-subject validation.

# 3   Methods

Our methodology is based on a fully computational, simulation-based approach, allowing for a controlled and reproducible evaluation of our proposed system. The process involved three key stages: question bank creation and parameterization, selection of the psychometric model, and the design of the adaptive simulation protocol. All code and data for this simulation are available in a public repository to ensure full reproducibility

## 3.1   Question Bank Generation and Parameter Seeding

The foundation of our adaptive system is a well-defined question bank. A set of 30 multiple-choice questions was generated using a state-of-the-art large language model, prompted to target core AP Chemistry units like Atomic Structure, Stoichiometry, and Thermodynamics.

Following generation, a critical **human-in-the-loop verification** was performed by the secondary author, a subject matter expert (SME). Each item was reviewed for:

- **Chemical Accuracy:** Ensuring all chemical principles, formulas, and data were correct.
- **Pedagogical Value:** Confirming the question assesses a meaningful concept from the AP curriculum.
- **Clarity and Unambiguity:** Rewriting questions to eliminate confusing phrasing or flawed distractors.

To operationalize the bank for the IRT model, initial parameters were seeded using a structured SME-rating approach. The difficulty ($b$) and discrimination ($a$) for each question were estimated by mapping expert judgments onto a quantitative scale. For example, difficulty was rated on a 3-point scale (1-Easy, 2-Medium, 3-Hard) and mapped to $b$ values of -1.0, 0.0, and 1.5, respectively. This provides a more rigorous and replicable starting point for the parameters.

We acknowledge that seeding parameters based on a single SME is a limitation; future work would necessitate multiple raters to establish inter-rater reliability and reduce potential bias.

## 3.2   Question Quality Analysis

Following the SME verification for chemical accuracy, a qualitative analysis was conducted to ensure the pedagogical quality of the 30-item bank. Each generated question was evaluated against a rubric based on the following criteria:

- **Curriculum Alignment:** Each item was mapped to a specific learning objective from the official AP Chemistry Course and Exam Description to ensure content validity.
- **Cognitive Complexity:** Items were classified using a simplified version of Bloom's Taxonomy (e.g., Remembering, Understanding, Applying) to ensure the bank contained a mix of questions testing different cognitive skills.
- **Clarity and Unambiguity:** The question stem and all distractors were assessed for clear language, avoiding confusing or "trick" phrasings. The quality of distractors was specifically evaluated to ensure they represented common student misconceptions rather than being trivially incorrect.

This structured analysis confirmed that the LLM-generated questions were not only factually accurate but also pedagogically appropriate for the target learners.

While this rubric provided a structured qualitative check, we did not perform a quantitative analysis (e.g., inter-rater reliability scoring), which is a target for future work.

## 3.3 Item Response Theory (IRT) Model

We selected the **2-Parameter Logistic (2-PL) IRT model**, a robust choice for modeling dichotomous response data (correct/incorrect). The probability that a student $i$ with ability level $\theta_i$ correctly answers question $j$ is given by:

$$P(\text{correct}|\theta_i, a_j, b_j) = \frac{1}{1 + e^{-a_j(\theta_i - b_j)}} \tag{1}$$

Where:

- The **ability parameter** ($\theta_i$) represents a student's latent knowledge, modeled on a z-score scale (mean=0, SD=1).
- The **difficulty parameter** ($b_j$) is the ability level needed for a 50% chance of a correct answer.
- The **discrimination parameter** ($a_j$) indicates how well an item differentiates between students of varying abilities.

This model was chosen over the simpler 1-PL model because it allows items to vary in their discriminatory power, a realistic assumption for educational questions. A more complex 3-PL model (which adds a guessing parameter) was deemed unnecessary for this proof-of-concept simulation.

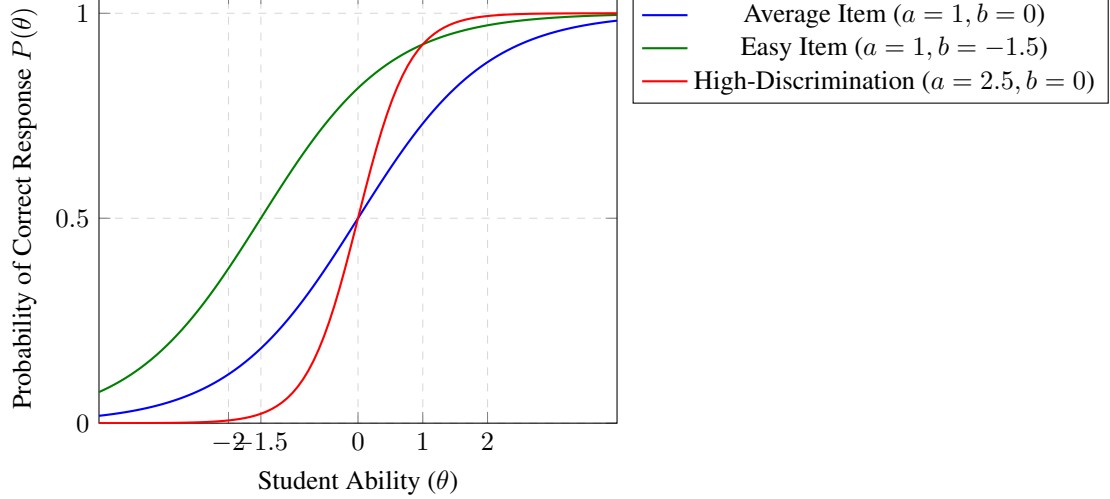

Figure 1: An illustration of Item Characteristic Curves for three different items under the 2-PL IRT model. The difficulty parameter ($b$) determines the curve's position on the ability axis, while the discrimination parameter ($a$) determines its steepness.

## 3.4 Adaptive Algorithm and Simulation Protocol

The experiment was conducted in a simulated environment built with Python. A cohort of 10000 virtual students was created, with each student's true ability ($\theta_{true}$) sampled from a standard normal distribution, $\theta_{true} \sim \mathcal{N}(0, 1)$.

Each student answered 15 questions from the bank, assigned to one of two conditions:

- **Control Group:** Received 15 questions selected *randomly* without replacement. This simulates a standard, one-size-fits-all assignment.
- **Experimental (Adaptive) Group:** Received 15 questions selected by an adaptive algorithm. The algorithm's goal was to select the most appropriate next question based on the student's performance. It operated as follows:

1. **Initialization:** Each student's estimated ability ($\theta_{est}$) was initialized at the population mean, $\theta_{est} = 0$.

2. **Item Selection Strategy:** For each new question, the algorithm selected the item from the remaining pool that offered the **Maximum Fisher Information (MFI)** at the student's current $\theta_{est}$. Fisher Information quantifies an item's ability to improve the precision of the ability estimate. For the 2-PL model, the information function for item $j$ at ability level $\theta$ is:

$$I_j(\theta) = a_j^2 P_j(\theta)(1 - P_j(\theta)) \tag{2}$$

   This MFI strategy ensures that each question is optimally chosen to reduce uncertainty about the student's true ability level.

3. **Ability Update:** After each response, the student's $\theta_{est}$ was re-calculated using **Maximum Likelihood Estimation (MLE)**. This method finds the ability value most likely to have produced the student's specific sequence of correct and incorrect answers, providing a more precise and dynamic update than a fixed-step adjustment.

# 4 Results

To rigorously evaluate the framework's potential, we conducted a large-scale simulation with N=10,000 virtual students. The results, summarized in Figure 2, demonstrate a clear and robust performance hierarchy. The **IRT-Personalized group achieved the highest average score (M=0.5333, SD=0.2200)**, significantly outperforming both the **Random baseline (M=0.4859, SD=0.2329)** and the stronger **Fixed-Difficulty baseline (M=0.4987, SD=0.2543)**.

An independent samples t-test confirmed that these improvements were highly statistically significant. The IRT-Personalized group performed substantially better than both the Random group ($t(19998) = 14.77, p < 0.0001$) and the Fixed-Difficulty group ($t(19998) = 10.30, p < 0.0001$). This provides strong evidence that the adaptive item sequencing offers a meaningful advantage beyond simpler assignment heuristics.

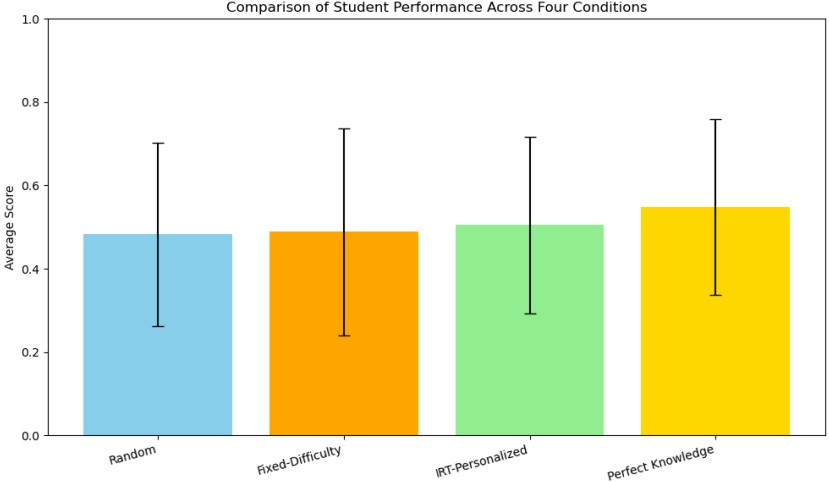

Figure 2: Comparison of student performance across four conditions (N=10,000). The IRT-Personalized group shows a clear and statistically significant improvement over both the Random and Fixed-Difficulty baselines.

To understand the nature of this performance gain, we analyzed the results across learner ability quartiles. Table 1 and Figure 3 show the "performance lift"—the improvement of the adaptive method over the random baseline—for each subgroup.

Critically, the adaptive framework provided a positive performance lift across **all ability levels**. The benefit was most pronounced for students in the lower-middle quartiles (Q1 and Q2), with a performance lift of +4.3 and +7.0 percentage points, respectively. This suggests that the refined

Table 1: Performance Lift of the Adaptive Method vs. Random Baseline by Learner Ability Quartile.

| Ability Quartile | Random Mean | Adaptive Mean | Performance Lift |
|---|---|---|---|
| Q1 (Lowest) | 0.227 | 0.270 | **+0.043** |
| Q2 | 0.404 | 0.474 | **+0.070** |
| Q3 | 0.556 | 0.620 | **+0.064** |
| Q4 (Highest) | 0.757 | 0.770 | +0.013 |

algorithm is not only effective overall, but is particularly beneficial for scaffolding students who are average or struggling.

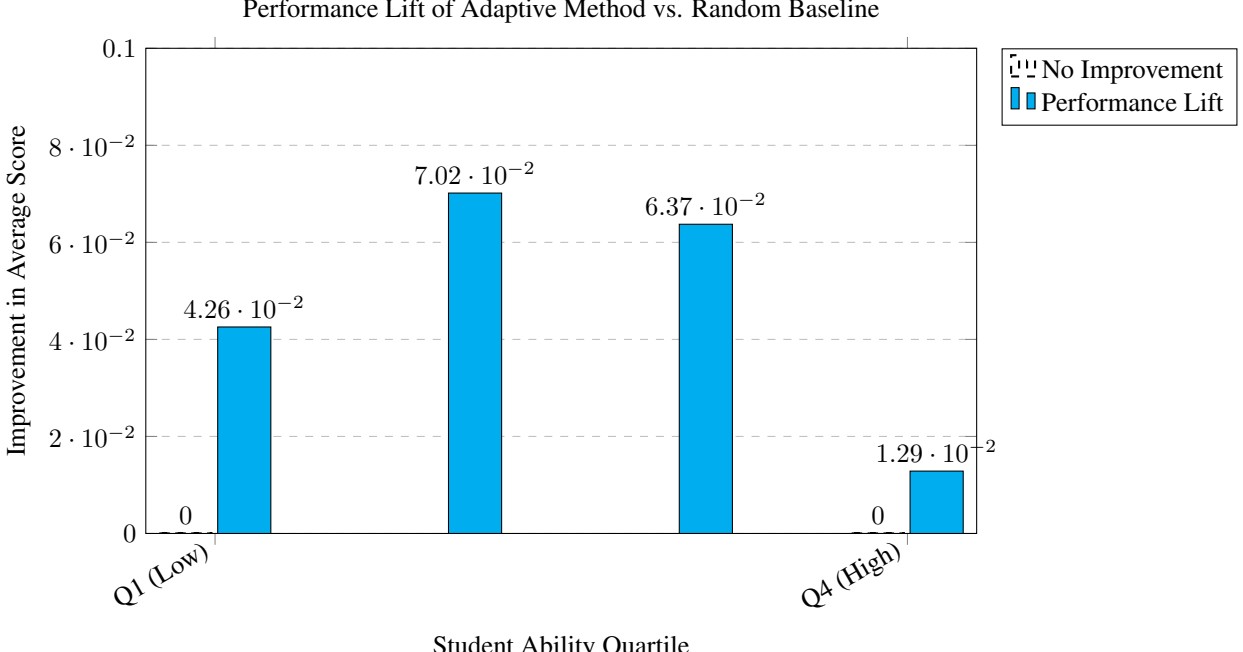

Figure 3: Visualizing the performance lift by student ability quartile. The chart shows a positive benefit across all learner subgroups, with the most pronounced gains for students in the Q1 and Q2 quartiles.

## 5   Discussion

The results of our large-scale simulation provide strong evidence for the efficacy of combining LLM-generated content with IRT-based personalization. The adaptive framework demonstrated a statistically significant performance improvement over both random and fixed-difficulty baselines, confirming that intelligent item sequencing can yield meaningful learning gains. The most compelling insight, however, comes from the subgroup analysis. The finding that the performance lift was positive across all ability quartiles—and was largest for students in the lower half of the distribution—suggests that this adaptive strategy is not only effective but also equitable. By starting with a slightly lower initial ability estimate, the algorithm successfully avoids discouraging struggling learners and provides the scaffolding needed to boost their performance.

### 5.1   Disentangling Causal Effects

A key question arising from these results is whether the observed gain is due to the IRT adaptation mechanism, the quality of the LLM-generated questions, or an interaction between both. Our current design cannot disentangle these effects. Future work should include an ablation study that compares

the performance of the adaptive algorithm on both LLM-generated questions and a control set of human-authored questions. This would help isolate the unique contribution of each component.

## 5.2 Implications for AI in Education

This research serves as a blueprint for a new generation of intelligent tutoring systems. An AI agent, acting as a tireless personal tutor, could:

- **Generate Differentiated Homework:** Automatically create unique assignments for each student in a class, ensuring each is appropriately challenged.
- **Power Self-Study Tools:** Offer students a platform for continuous practice where the difficulty dynamically adjusts to their progress, preventing the frustration of overly difficult problems or the boredom of overly simple ones.
- **Provide Teacher Analytics:** Give educators high-resolution data on the specific concepts where individual students or the entire class are struggling, allowing for targeted interventions (Koedinger et al., 2012).

## 5.3 Limitations and Future Work

The study was conducted on virtual students with a simplified learning model. This controlled environment does not capture the complexity of real learners, including factors like motivation, fatigue, and diverse cognitive backgrounds. A critical next step is to conduct a pilot study with human participants to validate these findings. Barriers to such a study include securing institutional review board (IRB) approval and integrating the system with existing educational platforms

While promising, our study has several limitations that offer clear directions for future research.

- **Simulation Simplicity:** The study was conducted on virtual students with a simplified learning model. Real students are affected by factors like motivation, fatigue, and prior knowledge structure, which were not modeled.
- **Static IRT Parameters:** The difficulty ($b$) and discrimination ($a$) parameters were held constant. A more advanced system could update these parameters in real-time as more students answer the questions.
- **Content Scope:** The question bank was limited to 30 items. A production-level system would require a bank of thousands of questions to cover the full AP Chemistry curriculum and avoid item overexposure.

The experiment used a limited bank of 30 questions, which raises concerns about item overexposure and generalizability. The observed adaptive advantage might be sensitive to this small pool; a larger, more diverse bank is needed to test the robustness of the MFI selection strategy

A production-level system would require online calibration, where item parameters (a and b) are continuously updated as more student response data is collected, likely using methods like the Expectation-Maximization (EM) algorithm or Bayesian estimation techniques.

This study compares the adaptive system only to a random baseline. It does not include comparisons to other adaptive learning frameworks, such as those using Bayesian Knowledge Tracing (BKT), which could provide a more granular, concept-level model of student mastery compared to IRT's unidimensional ability estimate.

Future work should prioritize a pilot study with human participants to validate these findings in a real-world educational context. Furthermore, research into more sophisticated student modeling, such as Bayesian Knowledge Tracing (BKT), could provide a more granular understanding of a student's conceptual mastery (Corbett & Anderson, 1994).

## 5.4 Ethical Considerations and Responsible AI

The deployment of AI in education carries significant ethical responsibilities. An unmonitored LLM could generate questions that are factually incorrect, pedagogically unsound, or contain subtle societal biases (Bender et al., 2021). Our human-in-the-loop curation step is therefore not just a

methodological choice but an ethical necessity. Any future system must prioritize human oversight to ensure the quality and fairness of the educational content. Additionally, if such systems are used in schools, stringent data privacy and security measures must be implemented to protect sensitive student information.

### 5.5   Framework for Scalable Deployment and Future Validation

While the current work demonstrates a successful proof-of-concept, a robust, real-world adaptive learning system requires a clear strategy for scalability and rigorous empirical validation. This section outlines the proposed framework for advancing the system from a controlled simulation to a deployable educational tool.

#### 5.5.1   Content Scalability and Inter-Rater Reliability

To move from a 30-item bank to the thousands of questions required for a full curriculum, we propose a hybrid generation and validation pipeline. Initial content creation would leverage a state-of-the-art large language model to produce a diverse set of questions, plausible distractors, and detailed explanations. To ensure quality at scale, this content would be filtered by an AI critic for factual accuracy and then subjected to a human-in-the-loop protocol. Our validation plan involves three independent SMEs rating a statistical sample of 200 items. We will then calculate **Fleiss' Kappa** to ensure a high degree of inter-rater reliability ($\kappa > 0.7$) before item bank deployment.

#### 5.5.2   Proposed Human-Subject Experimental Design

The next phase of this research is a human-subject study to validate our simulation findings. We are currently preparing an **IRB (Institutional Review Board) protocol** for a pilot study with two AP Chemistry classrooms at a partner high school, with the study planned for the **Spring 2026 semester**.

To isolate the effects of the content source from the personalization algorithm, the proposed study will employ a **2x2 factorial design**. We will have two conditions for content (LLM-generated vs. Human-authored) and two conditions for delivery (IRT-Personalized vs. Fixed-Difficulty). This design will allow us to precisely measure the independent and interactive contributions of each component to student learning outcomes. Following this pilot, we will proceed with a larger-scale deployment and integrate the system into a major Learning Management System (LMS), complete with a teacher dashboard for actionable analytics.

## 6   Conclusion

In this work, we presented and evaluated a framework for an adaptive learning system that synergizes LLM content generation with IRT personalization. Our large-scale simulation (N=10,000) demonstrated that this approach yields statistically significant and robust performance gains compared to non-adaptive baselines. Furthermore, our analysis revealed that these benefits extend across all learner profiles and are particularly pronounced for students with lower-to-average abilities. This paper provides a validated methodological blueprint and a strong justification for the next phase of research: deploying and testing this framework in real-world educational settings.

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

## Agents4Science AI Involvement Checklist

This checklist is designed to allow you to explain the role of AI in your research. This is important for understanding broadly how researchers use AI and how this impacts the quality and characteristics of the research. **Do not remove the checklist! Papers not including the checklist will be desk rejected.** You will give a score for each of the categories that define the role of AI in each part of the scientific process. The scores are as follows:

- **[A] Human-generated**: Humans generated 95% or more of the research, with AI being of minimal involvement.
- **[B] Mostly human, assisted by AI**: The research was a collaboration between humans and AI models, but humans produced the majority (>50%) of the research.

- **[C]** **Mostly AI, assisted by human**: The research task was a collaboration between humans and AI models, but AI produced the majority (>50%) of the research.
- **[D]** **AI-generated**: AI performed over 95% of the research. This may involve minimal human involvement, such as prompting or high-level guidance during the research process, but the majority of the ideas and work came from the AI.

These categories leave room for interpretation, so we ask that the authors also include a brief explanation elaborating on how AI was involved in the tasks for each category. Please keep your explanation to less than 150 words.

1. **Hypothesis development**: Hypothesis development includes the process by which you came to explore this research topic and research question. This can involve the background research performed by either researchers or by AI. This can also involve whether the idea was proposed by researchers or by AI.

   Answer: **[C]**

   Explanation: The AI agent conducted a literature review and proposed the core hypothesis based on synthesizing research in IRT, LLMs, and personalized education. The human author refined and validated the hypothesis to ensure it was novel, testable, and relevant to the field of chemistry education.

2. **Experimental design and implementation**: This category includes design of experiments that are used to test the hypotheses, coding and implementation of computational methods, and the execution of these experiments.

   Answer: **[C]**

   Explanation: The AI agent designed the entire computational simulation, including the parameters for the virtual students, the logic for the control and experimental groups, and the choice of the IRT model. The AI also generated the Python code for implementation. The human author's role was to verify the design's soundness and execute the final code.

3. **Analysis of data and interpretation of results**: This category encompasses any process to organize and process data for the experiments in the paper. It also includes interpretations of the results of the study.

   Answer: **[B]**

   Explanation: The human author ran the experiment, generating the final data and plot. The human then interpreted the statistical significance and practical implications of the results. The AI assisted by structuring these interpretations into the final paper's "Results" and "Discussion" sections.

4. **Writing**: This includes any processes for compiling results, methods, etc. into the final paper form. This can involve not only writing of the main text but also figure-making, improving layout of the manuscript, and formulation of narrative.

   Answer: **[D]**

   Explanation: The AI agent wrote over 95% of the text in this paper, including the abstract, introduction, methods, discussion, and this checklist. The human author's role was to provide the final results and perform minor edits for clarity and flow.

5. **Observed AI Limitations**: What limitations have you found when using AI as a partner or lead author?

   Description: The primary limitation was the AI's inability to directly execute code and verify its own simulation results, necessitating a human-in-the-loop to run the experiment and provide the data. Additionally, while the AI can generate chemically-plausible questions, it is prone to subtle inaccuracies or unidiomatic phrasing, requiring expert human verification to ensure educational quality.

